# Identification of Type-H-like Blood Vessels in a Dynamic and Controlled Model of Osteogenesis in Rabbit Calvarium

**DOI:** 10.3390/ma15134703

**Published:** 2022-07-05

**Authors:** Laurine Marger, Nicolas Liaudet, Susanne S. Scherrer, Nicolo-Constantino Brembilla, Olivier Preynat-Seauve, Daniel Manoil, Mustapha Mekki, Stéphane Durual

**Affiliations:** 1Biomaterials Laboratory, Division of Fixed Prosthodontics and Biomaterials, University Clinics of Dental Medicine, University of Geneva, 1 Rue Michel Servet, 1204 Geneva, Switzerland; laurine.marger@unige.ch (L.M.); susanne.scherrer@unige.ch (S.S.S.); mustapha.mekki@unige.ch (M.M.); 2Bioimaging Core Facility, University of Geneva, 1204 Geneva, Switzerland; nicolas.liaudet@unige.ch; 3Department of Pathology and Immunology, Faculty of Medicine, University of Geneva, 1204 Geneva, Switzerland; nicolo.brembilla@unige.ch; 4Department of Medicine, Faculty of Medicine, University of Geneva, 1204 Geneva, Switzerland; olivier.preynat-seauve@unige.ch; 5Division of Cariology and Endodontics, University Clinics of Dental Medicine, University of Geneva, 1204 Geneva, Switzerland; daniel.manoil@unige.ch; 6Division of Oral Diseases, Department of Dental Medicine, Karolinska Institutet, 171 77 Stockholm, Sweden

**Keywords:** bone regeneration, osteogenic mechanisms, osteogenic niches, type-H blood vessels, rabbit calvarial model

## Abstract

Angiogenesis and bone regeneration are closely interconnected processes. Whereas type-H blood vessels are abundantly found in the osteogenic zones during endochondral long bone development, their presence in flat bones’ development involving intramembranous mechanisms remains unclear. Here, we hypothesized that type-H-like capillaries that highly express CD31 and Endomucin (EMCN), may be present at sites of intramembranous bone development and participate in the control of osteogenesis. A rabbit model of calvarial bone augmentation was used in which bone growth was controlled over time (2–4 weeks) using a particulate bone scaffold. The model allowed the visualization of the entire spectrum of stages throughout bone growth in the same sample, i.e., active ossification, osteogenic activity, and controlled inflammation. Using systematic mRNA hybridization, the formation of capillaries subpopulations (CD31–EMCN staining) over time was studied and correlated with the presence of osteogenic precursors (Osterix staining). Type-H-like capillaries strongly expressing CD31 and EMCN were identified and described. Their presence increased gradually from the regenerative zone up to the osteogenic zone, at 2 and 4 weeks. Type-H-like capillaries may thus represent the initial vascular support encountered in flat bones’ development and which organize osteogenic niches.

## 1. Introduction

Angiogenesis is closely interconnected with tissue regeneration, as it provides the necessary vascular network for metabolically active tissue [1,2,3,4,5]. Even more so during bone regeneration, in which the lack of functional vascularization represents the main cause of failure [6,7,8,9]. However, successful bone regeneration does not only depend on sufficient vascularization but also on the appropriate cocktail of signals triggering incoming mesenchymal stem cells to differentiate into the osteogenic lineage [4]. As such, the interplay between ingrowing vessels and bone tissue has become a topic of scientific interest revealing the existence of a coupling between osteoblasts (OB) and endothelial cells (EC). OBs secrete vascular endothelial growth factor (VEGF) to promote angiogenesis [10]. In turn, ECs release several osteogenic factors such as fibroblast growth factors (FGF), platelet-derived growth factors (PDGFs), or bone morphogenetic proteins (BMPs) [11]. ECs are highly heterogeneous and specialized with respect to a function and a local microenvironment [12]. Recently, type-H vessels were discovered. Their main characteristic is a level of CD31 and Endomucin (EMCN) expression which is higher than that of other blood vessels. They are defined and analyzed on the basis of positive CD31 and EMCN expression. In osteogenic niches from the bone vascular growth front and the endosteum, type-H vessels were identified together with mesenchymal stem cells and osteoprogenitors (OP) [11]. A correlation exists between the concentration of OP and the presence of H-endothelial cells (HEC) and this particular type of vessel was shown to decrease with aging and loss of bone mass [11,13].

Humans display two independent mechanisms of bone growth and remodeling. Whereas endochondral ossification is responsible for long bone growth and regeneration [14], intramembranous ossification occurs in flat bones [15].

During endochondral ossification, chondrocytes stop proliferating to become hypertrophic, thereby forming a primary ossification center. Within this primary ossification center, secreted proangiogenic factors are responsible for inducing a vascular network invasion accompanied by ossification of the structure [14]. Type-H vessels are abundant in these osteogenic areas [11,16,17]. They are directly linked to arteriole columns distally interconnected [11] and surrounded by dense populations of osteoprogenitors [11]. H vessels are subsequently connected to the sinusoidal vasculature formed by another vessel subtype, termed L-type, poor in perivascular osteoprogenitors [18,19]. Co-expression of CD31 and Endomucin (EMCN) at high levels are characteristic of type-H endothelial cells, whereas expression levels remain low in L-type ECs [11].

This particular vascular network was not described precisely in the flat bones’ physiology, except in a single study that suggests the presence of type-H vessels in a calvarial bone regeneration model [20]. A vascular network resembling type-H capillary and spatiotemporally organized was also described in the calvarium but in a bioengineered model enriched in bone marrow stromal cells [21]. Intramembranous ossification shares mechanistic similarities with the endochondral process. Rather than chondrocytes condensation, a concentration of mesenchymal stem cells is observed to form an ossification center that creates at first a proangiogenic environment attracting the vascular network [15]. Such similitudes strongly suggest that H vessels, together with osteoprogenitors, could also play a central role in the development and regeneration of flat bones.

A better understanding of bone regeneration is key for improving bone scaffolds’ materials and designs.

In the present study, we postulated that type-H-like capillaries may be present at sites of intramembranous bone development and participate in the control of osteogenesis. We employed a calvarial bone augmentation model in the rabbit and aimed to control and visualize the entire process of bone growth over time. The formation of type-H-like capillaries was monitored over time and correlated with the presence of osteogenic precursors using systematic semi-quantitative mRNA hybridization for CD31, EMCN, and Osterix.

## 2. Materials and Methods

### 2.1. Experimental Design

The research described herein is part of a larger study characterizing several bone substitutes for which 37 animals were employed over a period of 12 weeks. We report here the results acquired with a single bone substitute at 2 and 4 weeks, which were obtained from 12 animals out of 37.

Four cylinders were screwed on rabbit skulls and filled with bone scaffolds. Two and four weeks after implantation, animals were sacrificed, samples were block sectioned before being processed for histology. Two staining protocols were performed on each biopsy (eosin hematoxylin and Masson’s trichrome) as well as in situ mRNA hybridization specific for CD31, Endomucin, and Osterix. Total bone synthesis, vascular characterization, and osteogenic precursors identification were performed with respect to time and height within the scaffolds (Figure 1A).

The following allocation table was used for the entire study: the substitutes were positioned at least once in each of the 4 cylinders placed on rabbits’ skulls at each time point. Furthermore, each animal received 4 different conditions.

The results we describe here were obtained with Bio-Oss (BO: Bio-Oss^®^, 250–1000 µm, 60% porosity, Geistlich Pharma AG, Wolhusen, Switzerland) at 2 and 4 weeks. Six biopsies were analyzed at both 2 and 4 weeks that represent a total of 12 biopsies obtained from 12 rabbits, according to the allocation table. In addition, we included controls for which the cylinders were filled with coagulated rabbit blood. Four controls were performed at 2 and 4 weeks, for a total of 8 biopsies (Table 1). These control samples were added to verify the pertinence of the model, i.e., that no ectopic bone growth could be observed without scaffold (Figure 2C, Appendix A, Figure A1).

### 2.2. Animals and Ethics

The protocol was approved by the local academic committee and the cantonal and federal veterinary agencies (authorizations GE/100/18).

Thirty-seven New Zealand white rabbits (male or female, 2.5–3 kg, older than 3 months, UNIGE breeding, Arare, Switzerland) were used in total. Twelve rabbits served in achieving data that we report in that study.

### 2.3. Surgeries

Surgical procedures were performed as previously described by our group [22,23]. Briefly, animals were left for one-week acclimation in the facility and received a prophylactic broad-spectrum antibiotic coverage (2 h preoperative, 3 days postoperative, enrofloxacin Baytril 10%, 5–10 mg/kg PO, Bayer, Leverkusen, Germany). On the day of the surgeries, they were pre-anesthetized by an injection of ketamine (25 mg/kg, 50 mg/mL, 0.5 mL/kg IM, Pfizer, New York, NY, USA) plus xylazine (3 mg/kg, 20 mg/mL, 0.15 mL/kg IM, Bayer). Deep anesthesia was induced by propofol 2% IV (Braun, Sempach, Switzerland) and maintained with sevoflurane 3% (Abbvie, Chicago, IL, USA) in pure oxygen. A remifentanil IV perfusion (Bichsel, Unterseen, Switzerland) (ear vein, 0.008–0.5 um/kg/min, 5 g/mL) ensured analgesia during the procedure. A midsagittal incision and periosteum elevation were then performed before screwing 4 PEEK cylinders and drilling 5 intramedullary holes (diameter 0.8 mm, depth ca. 1 mm) in the bone bed of each cylinder. After filling the cylinders with bone substitutes and closing (Figure 1B), the surgical site was sutured with intermittent non-resorbable sutures (Prolene 4.0, Ethicon, Somerville, NJ, USA). Animal received buprenorphine hydrochloride (Reckitt Benckiser, Slough, UK), every 6 h and up to 3 days (0.02 mg/kg, 0.03 mg/mL, 0.67 mL/kg SC) as postoperative analgesia. Sutures were removed after ca. 10 days.

Poly ether ether ketone (PEEK) cylinders and caps were produced by Boutyplast, Leyment, FR and fixed with CpTi Gr5 microscrews (diameter 1.2 mm, length 4 mm) from Global D, Brignais, FR. Bio-Oss was previously soaked in blood before placement, ca. 75 mg of particles were used per cylinder.

Animals were sacrificed at 2 and 4 weeks, the cylinders were removed, and the biopsies were fixed in formalin 4%. One month of decalcification (Osteosoft, Sigma, St Louis, MO, USA) was required before embedding the biopsies in paraffin. To ensure a thorough assessment of each biopsy, 3 levels of cuts were made from the middle of the biopsies, with a 150 µm offset. The sections were 4 µm thick.

### 2.4. Histological Analysis

Hematoxylin-eosin staining was employed to describe the general aspect of the section, of tissue and scaffold structure, and for cell identification and discrimination.

Masson–Goldner staining, which specifically labels collagen fibers green/blue, was employed to further discriminate cells from the surrounding connective tissue and for the semi-quantitation of newly formed bone. Observation was performed on a stereo video microscope (VHX-5000, Keyence, Itasca, IL, USA) following a procedure previously described by our group [22].

### 2.5. mRNA Hybridization and Image Analysis

With the aim of distinguishing capillaries subpopulations, mRNA in situ hybridization was used (ViewRNA^TM^ Tissue Assay Core Kit (Thermofisher, Watham, MA, USA) and the complementary blue module) according to manufacturer’s instructions. This assay is highly sensitive when compared to traditional fluorescent in situ hybridization (FISH). Twenty base pairs probes are used combined with a sequential branched DNA amplification technique to reach a single-copy sensitivity at single-cell resolution so that semi-quantitative analysis is allowed.

Three different probes were used: CD31 (PECAM1 type1, ref VF1-4240612, Thermofisher), Endomucin (EMCN type 6, ref VPKA3CN, Affymetrix), and Osterix (OSX type 6, Sp7 ref VF6-4237545, Affymetrix).

Two mRNA targets were detected by using simultaneously the type 1 and type 6 probe sets, that is CD31–EMCN and CD31–OSX. These 2 co-hybridizations were always performed on directly adjacent sections (4 µm thick) to allow subsequent digital merging of images thanks to morphological landmarks. The relative expression of CD31, EMCN, and OSX was therefore systematically analyzed on the same histological regions.

Briefly, sections were first deparaffinized, heated at 85 °C for 10 min, and digested with a protease solution (dil. 1/100) for 20 min at 40 °C. Once pretreated, the sections were re-fixed in 10% neutral buffer formalin for 5 min at room temperature. Probe sets were then applied for 3 h at 40 °C (CD31 (PECAM type 1)—Osterix (Osx type 6) and CD31 (PECAM type 1)—Osterix (Osx type 6)) to a final dilution of 1/80. As a negative control, probe set diluent was applied without any probe. Slices were rinsed vigorously several times before signal pre-amplifier and amplifier hybridization at 40 °C. Label probe (type 6) hybridization was then performed (dil. 1/1000, 15 min at room temperature) before the fast blue substrate was added (darkness, room temperature for 30 min). After rinsing, Label probe (type 1) hybridization was operated (dil. 1/1000, 15 min at 40 °C) and the fast red substrate was added for 60 min at room temperature. Slices were finally coverslipped.

Images were recorded under a fluorescence microscope (Zeiss Axio Imager 2, Oberkochen, Germany) at a 20-fold magnification, with the same intensity and exposure time. Each biopsy, (i.e., the whole cylinder content) was analyzed entirely, yielding an average of 80 captures digitally assembled. ViewRNA^TM^FastRed (λ_excitation_ = 553 nm and λ_emission_ = 568 nm) was used to detect CD31, ViewRNA^TM^FastBlue (λ_excitation_ = 653 nm and λ_emission_ = 668 nm) to detect EMCN and OSX.

Images of fluorescent mRNA hybridization staining of CD31 and EMCN were analyzed with QuPath 0.3.0 [24]. Each capillary was circled on a morphological basis combined with CD31 expression. The bone bed was also tagged to serve as the base for distance measurements. These identifications were always made at the same magnification, with an identical “selection brush” width. For each biopsy, 2 slides (in the center of the biopsy) were systematically analyzed twice by 2 operators. In total, 7255 capillaries were analyzed, i.e., an average of 269 per section. Within these delimited surfaces, mRNA punctate signals in each channel (green for CD31, red for EMCN) were segmented with a trained pixel classifier based on an artificial neural network. All these measurements were compiled in a custom-made analysis framework with a graphical user interface written in Matlab 2021b. Among other features, capillaries were characterized by their relative CD31 and EMCN areas and by their minimal distance to the bone bed. Thresholds to define high versus low CD31 and EMCN relative areas in capillaries were user-defined per sample. They were set in order to have most of the capillary density in the “CD31 Low/EMCN Low” category. Practically, a cross delimiting the different categories of capillaries with respect to their CD31–EMCN expression was placed on scatter plots. Capillaries’ density was color-coded, and the cross was always placed at the limit of the denser core, i.e., the red–orange–yellow capillaries’ population.

For the sake of clarity, we set the colors on our figures as follows: green for CD31 and red for EMCN in the images where only these two labels were visualized, green for CD31 and red for OSX in the images where only these two labels were visualized and finally green for CD31, red for EMCN and blue for OSX when the three labels were visualized on the same image.

### 2.6. Statistical Analysis

New bone volume data were checked for normal distribution and equivalence of variances. Unpaired *t*-tests were used to compare bone filling at 2 and 4 weeks. Significance levels were set to *p* ≤ 0.05.

Capillaries’ distance distribution was compared using a two-sample Kolmogorov–Smirnov test. Significance levels were set to *p* ≤ 0.05.

## 3. Results

### 3.1. Clinical Course, Biopsies Macroscopical Assessment

All rabbits survived surgery as well as the postoperative period of bone growth (up to 4 weeks). None displayed signs of infection or inflammation at the implantation site. Biopsies also appeared to be devoid of inflammatory signs. In samples collected 2 weeks post-implantation, the upper part of the biopsy was sometimes lost, since the particles were obviously not yet colonized by regenerative cells from ca. the middle of the cylinder in height. Therefore, the cohesion of the particles at the top of these biopsies was insufficient to resist the frictions imposed by cylinder removal. At 4 weeks, the biopsies were more compact, suggesting complete colonization of the scaffolds. The sham sites did not support bone growth in height (Figure 2C), yet very little amounts of newly formed bone were observed at the bone bed surface, coming from the healing of the intramedullary hole (Appendix A, Figure A1).

### 3.2. The Dynamic and Controlled Model of Osteogenesis, Histology and Histomorphometry

The model was calibrated in terms of the amount of scaffold-forming particles inside the cylinders. These particles were evenly distributed throughout the volume of the cylinder.

At each time point, three distinct tissue zones were successively visible along the height of the scaffold (Figure 2 and Figure 3). From the bone bed up, one distinguished: 1—a zone of bone growth and remodeling connected to the bone bed (RZ), 2—an osteogenic zone (OZ), richly vascularized and essentially composed of osteoid tissue, and 3—a granulation zone (GT) characterized by the presence of inflammatory cells and a strong vascularization.

GT migrated steadily from the bone bed 3 days after implantation (data not shown) to the middle of the scaffold at 2 weeks and reached its top at 4 weeks. The osteogenic zone, extending on a region of about 2 mm height at both 2 and 4 weeks, directly followed this upward migration of GT, followed by the remodeling zone, that increased in size and height from about 1 to 3 mm between 2 and 4 weeks (Figure 2A,B and Figure 3A,B).

All stages of bone regeneration were therefore represented on every single biopsy. The corresponding tissue zones migrated and evolved over time, depicting a dynamic model of osteogenesis, controlled by the scaffold.

Granulation tissue: red blood cells from the coagulum, in which the particles were mixed at implantation, were largely observed at 2 weeks and to a lesser extent at 4 weeks (Figure 2A,B). Below, the granulation tissue showed a significant proportion of capillaries at 2 and 4 weeks. This relatively homogeneous tissue surrounded the particles and consisted mainly of mononuclear and some polymorphonuclear cells. Osteogenic precursors expressing OSX were not observed in this area at 2 weeks (Figure 3A.3), with very few cells visible after 4 weeks (Figure 3B.3).

Osteogenic zone: the granulation tissue evolved progressively into a highly vascularized osteoid tissue (Figure 2A,B). The lower towards the bone bed, the more mineralized the fibrous matrix appeared. In its lowest part, osteoblasts could be seen organizing themselves in rows to form new bone tissue in direct contact with the particles. In this osteogenic zone, there was a very high expression of OSX, which demonstrates the presence of a large number of bone precursors (Figure 3A.1,B.1).

Remodeling zone: a progressive evolution from the osteogenic zone to an increasingly mature bone tissue closer to the bone bed was observed (Figure 2A,B). The particles were largely osteointegrated and the newly formed bone displayed trabeculae, themselves lined by rows of osteoblasts. As early as 2 weeks, and more importantly at 4 weeks, the lamellar bone could also be observed. Between the trabeculae and the particles, a loose and highly vascularized connective tissue was visible. Osteogenic precursors expressing OSX were found in this zone, mainly within the loose connective tissue, close to the bone growth zones, and in a lower proportion than in the osteogenic zone both at 2 and 4 weeks (Figure 3A.2,B.2).

The remodeling zone shifted upwards over time, thereby increasing the proportion of bone filling in the cylinder from 1.4% ± 0.3 at 2 weeks to 4.7% ± 1 at 4 weeks (% total area within the cylinder) (Figure 2C). The mean height of the bone tissue with respect to the bone bed increased from 0.7 ± 0.3 mm at 2 weeks to 2.4 ± 0.6 mm at 4 weeks. During the same time span, the mean height of the granulation tissue migrated from 2.5 ± 0.2 mm to 4.2 ± 0.2 mm.

### 3.3. Different Capillaries Subsets May Be Assessed with Respect to Their CD31–EMCN Relative Expression

Each biopsy was analyzed on its entire surface after mRNA hybridization with specific probes for CD31 and EMCN mRNA. Based on morphology and the level of CD31 expression, each capillary was circled (Figure 4A,C). The relative CD31 and EMCN areas in these capillaries were assessed and plotted on dot plots where each dot represented one capillary (Figure 4C, right). There was a differential expression of CD31 and EMCN in the capillaries within the biopsies analyzed. In the representative example shown in Figure 4B,C, a group of three contiguous capillaries was analyzed. The capillary labeled “3” expressed CD31 and EMCN at higher rates than capillaries “1” and “2”.

Based on these dot plots, we further investigated the repartition of these different capillaries’ subsets, especially the capillaries co-expressing high levels of CD31 and EMCN.

### 3.4. Capillaries Expressing the Highest Levels of CD31 and EMCN Are Preferentially Located within the Osteogenic Zone

When sorting all the capillaries observed at 2 and 4 weeks according to their relative expression of CD31 and EMCN, we could define a population co-expressing high levels of CD31 and EMCN (“CD31 High/EMCN High”). These were, however, under-represented compared to the main population defined as “CD31 Low/EMCN Low”. Furthermore, one observed a slight increase in EMCN expression levels that was accompanied by a slight decrease in CD31 expression levels in the general capillaries’ population between 2 and 4 weeks (Figure 5A).

The “CD31 Low/EMCN Low” main population was used as a reference in further analysis.

“CD31 High/EMCN High” capillaries were observed from 0 to ca. 3 mm in height at 2 weeks. They were mainly observed between 1 and 2 mm, their presence decreasing beyond 2mm. At 4 weeks, the main zone in which these “CD31 High/EMCN High” capillaries were observed had migrated between 2 and 4 mm. In contrast, the “CD31 Low/EMCN Low” capillaries were mostly observed in close vicinity to the bone bed at 2 weeks, their probability of being observed decreasing gradually when approaching the 3 mm mark. At 4 weeks, most “CD31 Low/EMCN Low” capillaries were still located within the first millimeter near the bone bed, and appeared homogenously distributed, albeit less abundant, higher up (Figure 5B and Figure 6A).

By superimposing these distribution graphs with the mean heights of bone and GT, we noted that “CD31 High/EMCN High” capillaries were predominantly distributed between the new bone and the GT at both 2 and 4 weeks. By contrast, “CD31 Low/EMCN Low” capillaries, although present throughout the whole regeneration area, were predominantly present within the bone tissue at 2 and 4 weeks (Figure 5C).

Finally, a representative biopsy was observed for which the capillary distribution was plotted on a histological image. The density of the “CD31 High/EMCN High” population started to gradually increase from the remodeling zone to reach its highest density in the osteogenic zone, between bone and GT, at 2 and 4 weeks. An inverse pattern was observed for the “CD31 Low/EMCN Low” capillary population (Figure 5D).

Around the “CD31 High/EMCN High” capillaries of the OZ, we observed high concentrations of osteogenic precursors expressing OSX, at 2 and 4 weeks (Figure 6A,A.1,B.1). These high concentrations of “CD31 High/EMCN High” capillaries and osteogenic precursors could be described as osteogenic foci. In the remodeling zone, CD31 High/EMCN High and osteogenic precursors were also present in connective and as yet unmineralized tissues, albeit in smaller proportions and more diffusely distributed. In the granulation tissue, “CD31 High/EMCN High” capillaries were rare and osteogenic precursors were absent at 2 weeks (Figure 6A,A.3). More “CD31 High/EMCN High” were observed at 4 weeks, with the emergence of few surrounding osteogenic precursors (Figure 6A,B.3).

## 4. Discussion

In this study, we employed a vertical bone augmentation model that allowed spatiotemporal control over the osteogenesis process. This model enabled the analysis of the entire spectrum of stages throughout bone growth on the same sample over time. Furthermore, mRNA in situ hybridization demonstrated that CD31- and EMCN-expressing capillaries, typical of type-H vessels found during long bone growth, are also present during intramembranous osteogenesis.

In the bone regeneration model employed herein, several identical cylinders are screwed onto the skull of an animal and filled with a scaffold [22,23]. The scaffold is necessary for bone growth, as it triggers a vertical “ectopic” osteogenesis that extends above the native calvarial bone plateau. The regenerative cells need to be guided and supported for new bone tissue to be synthetized. The choice of the scaffold is therefore crucial, both in its biochemical and architectural nature. This model may therefore be considered as an extension of the bone system in which the tissue growth is controlled and may advantageously be visualized.

In the present study, the sham samples were filled with coagulated blood only. In these controls, no bone synthesis was observed apart from the healing of intramedullary perforations created to attract osteogenic precursors and vascularization. This is in line with previous studies that also relied on such an ectopic augmentation model [22,23,25,26]. The scaffold employed consisted of 0.25–1 mm bovine bone particles that have represented a gold standard in orofacial bone regeneration for several decades. The particles are compacted to create an interconnected scaffold that displays around 60% porosity, which is ideal for vascular and bone growth [27].

Additionally, the choice of the animal model is also important. We relied on rabbit skulls, a model that allows the simultaneous assessment of four cylinders [22,23,28], and thus the testing of four simultaneous conditions on the same animal. Such optimization of the workflow constitutes an important ethical argument.

Finally, the morphology and bone metabolism of this model are similar to humans [29], which makes it a clinically relevant model, used in ca. 80% of calvarial studies of vertical augmentation. The calvarial model may be clinically translated as a “one-wall defect”, such as a class IV defect in the jaw. The stringency of this model permits the accurate assessment of the vertical osteoconduction of the materials that are being evaluated. Herein, we advantageously employed the model’s characteristics to scrutinize the cellular and vascular mechanisms at play during the regeneration process.

The process of bone augmentation reproduces the mechanisms of bone growth and regeneration. The site is first colonized by inflammatory cells, particularly neutrophils and macrophages, which will first form a matrix within the coagulum that will evolve into granulation tissue. This tissue attracts vascular and osteogenic cells by chemotaxis. A new bone matrix will therefore be synthesized, mineralized, and remodeled. Our calvarial model advantageously allows the concurrent analysis of these stages on a single biopsy. We have thus described three successive zones starting from the bone bed: first, a remodeling zone, second, an osteogenic zone, and third, a granulation tissue. These zones shift over time to ultimately fill the entire volume with mature bone. Using a hybrid collagen–particulate bone substitute in the same model, we have previously shown that the filling was satisfactory and almost complete after 12 weeks [22]. In the present study, we used a different scaffold, i.e., the gold standard Bio-Oss^®^, and we purposely limited the harvesting time points to the first 4 weeks of the process, during which osteogenesis is most important. The bone filling was ensured to arrive at 4 weeks at a level of 2.4 mm average height and a filling of 4.7%. These measures fit with data from previous studies [26] and with our own data [22], where ca. 12% bone filling over the entire scaffold was reached after 12 weeks. With 50% filling at 4 weeks and about 5% new bone, we can extrapolate to 10–12% at 12 weeks with a complete cylinder filling and consider the bone growth process in accordance with the model we used.

The remodeling zone increased in size from 2 to 4 weeks as bone filling augmented. At the same time, the osteogenic zone also migrated upwards, attracted by the granulation tissue that reached the top of the scaffold. The osteogenic zone maintained its dimensions in height, approximately 2 mm, while shifting towards the top of the scaffold. It is worth mentioning that the characterization of this osteogenic zone, which is essentially rich in osteogenic precursors, was not only performed on a histological and morphological basis, but also relied on the molecular identification of Osterix using the mRNA in situ hybridization technique that we developed. In addition, the three zones were richly vascularized.

Considering all these observations, this model is a dynamic model in which bone growth, and thus osteogenesis, is controlled in space and time.

We further developed and validated an in situ mRNA hybridization technique to evaluate the expression of CD31 and EMCN in capillaries and Osterix in osteogenic precursors. It was technically impossible to perform a triple CD31–EMCN–OSTERIX hybridization. Nevertheless, we were able to systematically perform CD31–EMCN and CD31–Osterix co-hybridizations on directly adjacent sections (4 µm thickness), so that photomicrographs could easily be superimposed thanks to morphological landmarks, (e.g., the particles). We were thus able to validate the presence of certain capillaries surrounded by osteogenic precursors.

From a technical standpoint, each section was analyzed by two specialists who circled the capillaries following the same methodology. A significant number of capillaries were identified and analyzed for each biopsy. Finally, we developed an analytical tool that enabled us to demonstrate that the capillaries identified in our biopsies express CD31 and EMCN (we verified the translation of the CD31 and EMCN glycoproteins in tissues by immunofluorescence (data not shown)). Moreover, the technique was sensitive enough to show variations in the expression of these two markers, and thus to analyze subpopulations, notably the “CD31 High/EMCN High” capillaries.

In the long bone development, the process of bone growth is termed endochondral. During this process, type-H capillaries that strongly express CD31 and EMCN are described in direct contact with osteogenic precursor concentrations [11,16,17]. In the intramembranous bone growth process encountered in flat bones from the orofacial area, mesenchymal stem cells concentrate and differentiate into bone cells instead of chondrocytes in the endochondral process. Irrespective of the process, the osteogenic precursors are therefore concentrated in foci ultimately leading to bone synthesis. In view of these similarities, it is therefore highly likely that the intramembranous process has a vascular system similar to that found in the endochondral process for maintaining osteogenic niches.

Our first observation showed that all the tagged capillaries expressed CD31 and EMCN and were distributed over the entire surface analyzed, that is in all tissue areas. We then focused our analyses on the capillaries that expressed the highest levels of CD31 and EMCN. By mapping these “CD31 High/EMCN High” capillaries, we observed that they were present in all the regions at 2 weeks, with a preferential distribution above the new bone tissue, in a zone that we qualified as osteogenic since i. it concentrated a high proportion of osteogenic precursors and ii. its collagenous matrix was beginning to mineralize. The further away from the bone bed and the remodeling zone, the more these capillaries were present. This profile was also observed at 4 weeks, but shifted upwards, as the remodeling zone had grown in height. The main population of capillaries expressed the lowest levels of CD31 and EMCN and was labeled “CD31 Low/EMCN Low”. Their distribution displayed an inverted profile when compared to “CD31 High/EMCN High” capillaries, with a majority of capillaries present in the remodeling zone.

Around the “CD31 High/EMCN High” capillaries of the osteogenic zone, foci of osteogenic precursors were concentrated, both at 2 and 4 weeks. Interestingly, we could also observe osteogenic precursors in the remodeling zone, albeit in a lower proportion and more diffusely distributed than in the osteogenic zone. Specifically, these precursors were localized in the fibroconnective tissue, where “CD31 High/EMCN High” were also observed, and not in direct contact with trabeculae or newly synthesized osteons. Within the remodeling zone itself, it seems that osteogenic precursors could still be slightly mobilized in the regions not yet mineralized, in the vicinity of “CD31 High/EMCN High” capillaries.

In light of these analyses, there appears to be a population of H-like capillaries strongly expressing CD31–EMCN and promoting the migration and concentration of osteogenic precursors in areas adjacent to active bone growth.

The resemblance between the endochondral and intramembranous systems does not end with this observation. Indeed, in the endochondral system, it was shown that type-H capillaries decrease with age, especially in embryonic bone development, as ossification increases [11,13]. Our spatiotemporal model of bone growth depicts a time-dependent vertical migration of tissue zones that reproduces the temporal evolution of bone development. Type-H-like capillaries decrease proportionally with time in mineralized areas to increase in adjacent as yet unmineralized areas. Biopsies were not analyzed over periods longer than 4 weeks herein. That said, current results may further suggest that the “CD31 High/EMCN High” capillary population should decrease over the entire height of the cylinder and that the relative CD31–EMCN expression dot plots would be more homogeneous, with a population centered on the dense “CD31 Low/EMCN Low” core.

It is further interesting to note that the expression of EMCN in the main capillary population increases slightly in time, unlike CD31 expression. EMCN is strongly involved in neovascularization [30], while CD31 plays an important role in intercellular junctions and the mechanisms of leukocyte transendothelial migration [31,32], especially for neutrophils and monocytes. When CD31 decreases, the migration of these cells through the capillaries to the tissues decreases [33]. We observed that “CD31 Low/EMCN Low” are essentially present in the remodeling zone, where angiogenesis is extremely important, and neutrophils and macrophages are much less active than in granulation tissue and osteogenic zones. Increased expression for EMCN and decreased expression for CD31 could therefore be explained by the physiological and metabolic stage of the tissues in which these capillaries are found.

In conclusion, we identified and mapped capillaries that express high levels of CD31 and EMCN in an intramembranous bone growth model. Because these capillaries highly resemble the type-H vessels typical of endochondral bone development, we labeled them herein as type-H-like vessels. Such type-H-like vessels appeared preferentially located in osteogenic zones adjacent to bone growth and remodeling areas and were surrounded by foci of osteogenic precursors. Whereas further histological and molecular characterization is warranted, these capillaries may represent the central structure of the osteogenic niche of developing or regenerating flat bones. In time, it may be possible to qualitatively and quantitatively improve bone growth by controlling and promoting the development of these capillaries. One key factor for the control of this capillary system may lie in the fine-tuning of the architecture and composition of bone substitutes.

## Figures and Tables

**Figure 1 materials-15-04703-f001:**
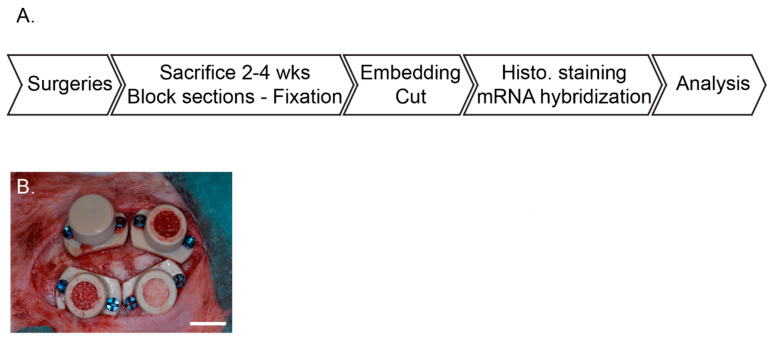
Study overview (**A**) main study milestones. (**B**) Representative picture of the cylinder placement filling and closure on the rabbit calvarium.

**Figure 2 materials-15-04703-f002:**
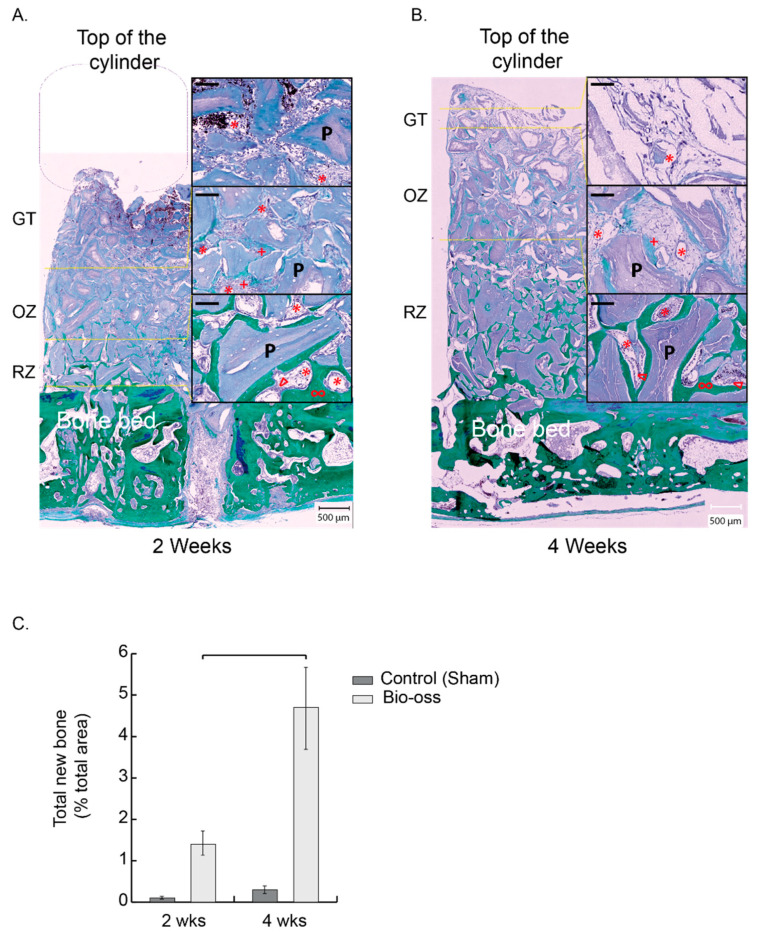
Description of the dynamic and controlled model of osteogenesis. (**A**,**B**) New bone development in a representative entire cylinder content filled with BO at 2 weeks (**A**) and 4 weeks (**B**): Left panels: whole cylinder content—Right panels: magnifications from the delimited zones RZ, OZ, and GT (yellow dotted lines). RZ is the remodeling zone in which new bone trabeculae (green, ∞) surround the particles (purple, P) and are lined by osteoblast rows (∆) and a rich vascularized (*) fibroconnective tissue. OZ is the osteogenic zone, a fibroconnective tissue and an osteoid tissue, highly vascularized, in which the matrix is being mineralized and appears in light green (+). GT is the granulation tissue that migrates to the top of the scaffold over time, dragging the OZ and the RZ. At 2 weeks, a missing part from the scaffold, not colonized by cells, was detached and is figured by a purple rounded square. Black bars in the magnifications: 100 µm. (**C**) Total new bone measured in the whole cylinder (% total area) at 2 and 4 weeks. Bio-Oss n = 6 at 2 and 4 weeks (dark grey), control (sham) n = 4 at 2 and 4 weeks (light grey), results expressed as mean +/− sem. Bracket *p* < 0.05.

**Figure 3 materials-15-04703-f003:**
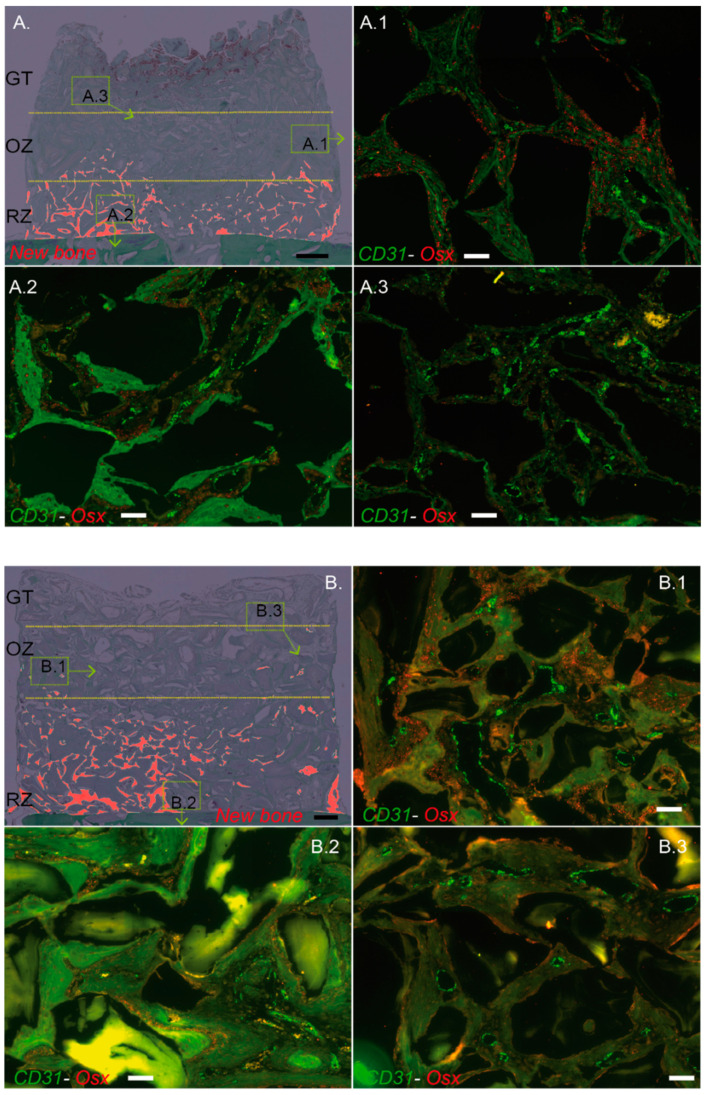
Description of the dynamic and controlled model of osteogenesis. Cylinder contents pictures from Figure 2A,B were digitally processed to highlight the new bone in red ((**A**), 2 weeks; (**B**), 4 weeks). RZ, OZ and GT are delimited by yellow dotted lines. A representative zone from OZ (**A.1**,**B.1**), RZ (**A.2**,**B.2**), and GT (**A.3**,**B.3**) was magnified and analyzed by RNA hybridization for CD31 (green dots) and Osterix (OSX, red dots). Note that each zone was highly vascularized either at 2 or 4 weeks, as assessed by the green dots tagging specifically CD31 mRNA (**A.1**–**3**,**B.1**–**3**). Osteogenic precursors in which OSX is highly expressed (red dots, OSX mRNA) were mostly observed in the osteogenic zone at 2 (**A.1**) and 4 weeks (**B.1**). OSX precursors were also present in the remodeling zone but at a lower rate when compared to OZ, either at 2 or 4 weeks (**A.2**,**B.2**), respectively, whereas they were very sparsely observed within the GT (**A.3**,**B.3**). Black bars: 500 µm, White bars: 100 µm.

**Figure 4 materials-15-04703-f004:**
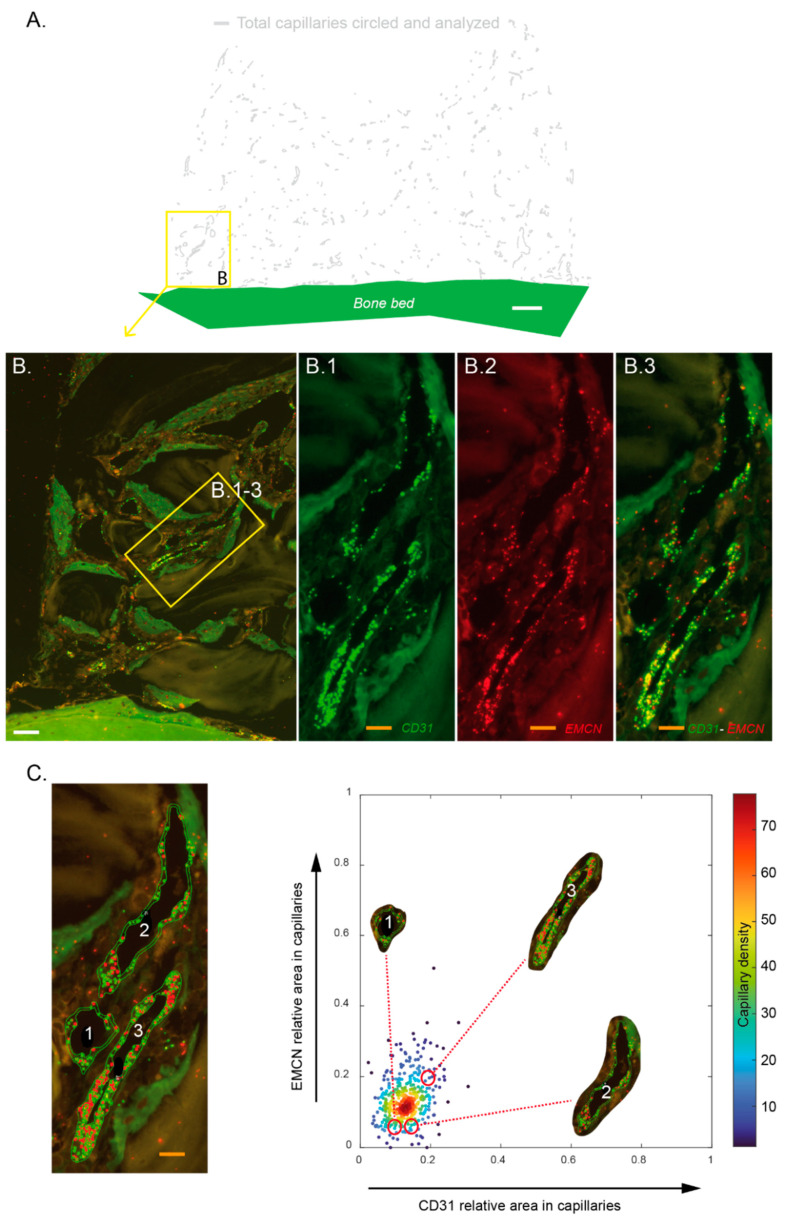
Assessment of the relative CD31 and EMCN expression in capillaries. Each sample (n = 6 at 2 and 4 weeks) was hybridized with specific probes targeting CD31 and Endomucin (EMCN) mRNA. Resulting slides were entirely scanned by using a fluorescent microscope. Based on morphology and CD31 staining, the whole capillaries were circled and the relative areas of CD31 dots (green)—EMCN (red) within these unitary perimeters was calculated. (**A**) Representative image of a cylinder’s entire content in which capillaries that were circled are shown on a digital projection that excludes tissues and particles to facilitate observations. White bar: 500 µm (**B**) magnification of a zone from (**A**) in which 3 contiguous capillaries are observed and were magnified to visualize CD31 mRNA expression (**B.1**), EMCN mRNA expression (**B.2**), and CD31–EMCN mRNA coexpression (**B.3**). White bar: 100 µm, orange bars: 25 µm. (**C**) Left: the 3 capillaries from (**B**) were circled and numerically tagged to facilitate their identification. Orange bar: 25 µm; right: scatter plot of EMCN vs. CD31 relative areas. Each point depicts a capillary belonging to the image shown in Figure 4A. The capillaries are color-coded according to their probability density (smoothed by a kernel density estimator for multivariate data) of having a given CD31 and EMCN relatives areas. These three capillaries show differential CD31–EMCN expression: capillary n°3 exhibits a higher CD31 and EMCN relative areas than the two others.

**Figure 5 materials-15-04703-f005:**
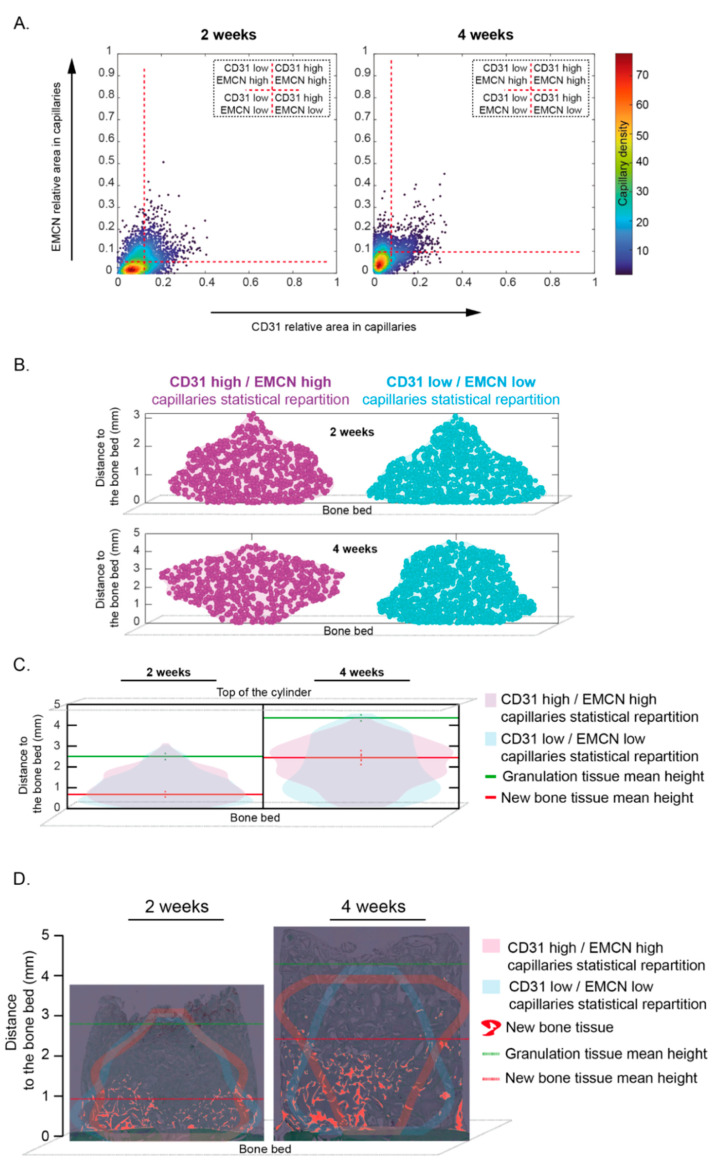
Capillaries subsets distribution within the cylinder contents at 2 and 4 weeks. (**A**) Scatter plots of EMCN vs. CD31 relative areas within capillaries for the whole samples at two weeks (**left**, n = 6) and at four weeks (**right**, n = 6). Each point is a color-coded capillary for its probability density of having such an EMCN and CD31 relative areas. Horizontal and vertical red dashed lines indicate, respectively, the EMCN and CD31 thresholds used to classify the capillaries in four categories. Thresholds were set in order to have most of the capillary density in the “CD31 Low/EMCN Low” category. (**B**) “Violin plots” representing the capillaries’ distribution within the whole cylinder content at two weeks (n = 6, **top**) and four weeks (n = 6, **bottom**). Purple and blue colors indicate, respectively, the “CD31 High/EMCN High”, and “CD31 Low/EMCN Low” capillaries and the horizontal spreading indicates the probability density of detecting capillaries at a given distance (smoothed by a kernel density estimator). (**C**) “Violin plots” representing the “CD31 High/EMCN High” (purple) and “CD31 Low/EMCN Low” (blue) capillaries distribution within the whole cylinder content, from the bone bed to the top of the cylinder, superimposed at 2 weeks (**left**, n = 6) and 4 weeks (**Right**, n = 6, **top**). Granulation tissue (green) and new bone tissue (red) mean height +/− sem (dotted vertical lines) were added. Note that the highest proportion of “CD31 High-EMCN High” capillaries was observed in-between the new bone tissue and the granulation tissue, namely within the osteogenic zone, either at 2 or 4 weeks. (**D**) “Violin plots” contours representing the “CD31 High/EMCN High” (purple) and “CD31 Low/EMCN Low” (blue) capillaries distribution of a representative sample at 2 and 4 weeks were superimposed to the corresponding cylinder contents pictures at 2 and 4 weeks. The same representative sample was conserved from Figure 2A and Figure 3A at 2 weeks and Figure 2B and Figure 3B at 4 weeks. The newly formed bone was highlighted in red. Granulation tissue (green) and new bone tissue (red) mean heights were added. Most of “CD31 High-EMCN High” capillaries were observed in-between the new bone tissue and the granulation tissue, within the osteogenic zone, either at 2 or 4 weeks. On the other hand, the largest proportion of “CD31 Low-EMCN Low” capillaries was observed in the new bone tissue, within the remodeling zone. Distance distribution was compared using a two-sample Kolmogorov–Smirnov test: at two weeks *p*-value~0.008 and at four weeks *p*-value << 0.001. Thus, the distance from the bone bed to the “CD31 High/EMCN High” and “CD31 Low/EMCN Low” capillaries appear not to follow the same continuous distribution either at 2 or 4 weeks.

**Figure 6 materials-15-04703-f006:**
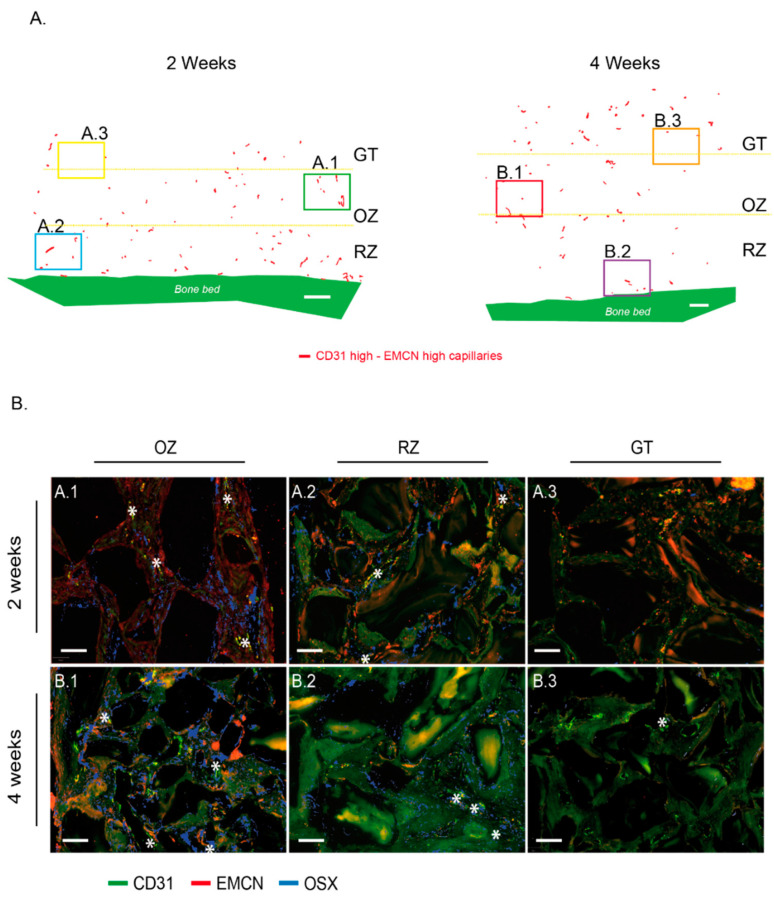
“CD31 High/EMCN High” capillaries distribution within the cylinder contents at 2 and 4 weeks and correlation with the presence of Osterix-expressing osteogenic precursors. (**A**) CD31 High/EMCN High capillaries from a representative image of a cylinder’s entire content at 2 weeks (**left**) and 4 weeks (**right**) are shown on a projection where tissues were digitally removed. The same representative sample was conserved from Figure 2A and Figure 3A, and Figure 5D at 2 weeks, Figure 2B and Figure 3B, and Figure 5D at 4 weeks. Granulation tissue (GT), osteogenic zone (OZ), and remodeling zone (RZ) were delimited by yellow dotted lines. White bar: 500 µm. (**B**) Representative magnifications from (**A**)—cylinder content at 2 weeks: OZ (**A.1**), RZ (**A.2**), and GT (**A.3**) and (**B**)—cylinder content at 4 weeks: OZ (**B.1**), RZ (**B.2**), and GT (**B.3**). Samples were analyzed by RNA hybridization for CD31 (green), EMCN (red), and OSX (blue). The “CD31 High/EMCN High” capillaries are tagged by a white star. Note that OSX-expressing osteogenic precursors were mainly observed in the osteogenic zone, in the vicinity of “CD31 High/EMCN High” rich regions, both at 2 (**A.1**) or 4 weeks (**B.1**). Despite the presence of “CD31 High/EMCN High” capillaries in the RZ, such concentrations of osteogenic precursors were not observed at 2 (**A.2**) or 4 weeks (**B.2**). A very weak expression of OSX was observed in the GT zone at 4 weeks (**B.3**), suggesting that osteogenic precursors migrate towards the region where “CD31 High/EMCN High” capillaries were observed. White bars: 100 µm.

**Table 1 materials-15-04703-t001:** Samples.

	2 Weeks/6 Animals	4 Weeks/6 Animals
Bio-Oss (BO)	N = 6	N = 6
Sham (coagulated blood)	N = 4	N = 4

## Data Availability

The data that support the findings of this study are available from the corresponding author, upon reasonable request.

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
