# Peer review of "Identification of Type-H-like Blood Vessels in a Dynamic and Controlled Model of Osteogenesis in Rabbit Calvarium"

_materials, 2022, doi:10.3390/ma15134703_

Round 1

Reviewer 1 Report

We congratulate the Authors for their work nonetheless It is difficult to draw important conclusions for the human from this animal model.

Author Response

We congratulate the Authors for their work nonetheless It is difficult to draw important conclusions for the human from this animal model.

> We thank the reviewer for her-his quite positive assessment of our work.

Concerning the conclusion drawn from this model to human clinic, we discussed this point in the first paragraph of discussion. Since it appeared that the relation to human clinic was not enough explained, we added this highlighted  text to the first paragraph of the discussion:

In the bone regeneration model employed herein, several identical cylinders are screwed onto the skull of an animal and filled with a scaffold [22, 23]. The scaffold is necessary for bone growth, as it triggers a vertical "ectopic" osteogenesis that extends above the native calvarial bone plateau. If the regenerative cells are not guided and supported, then no bone tissue is naturally synthesized. The choice of the scaffold is therefore crucial, both in its biochemical and architectural nature. In the end, this model is an extension of the bone system in which the tissue growth is controlled and may be easily visualized, a window on the bone metabolism.

In the present study, the Shams were filled with coagulated blood only. Apart from the healing of intramedullary perforations made to attract osteogenic precursors and vascularization, no bone synthesis was observed. This is in accordance with other studies in which this model of ectopic augmentation was used [22,23,25,26]. The scaffold employed consisted in 0.25-1 mm bovine bone particles that represent a gold standard in orofacial bone regeneration for several decades. After compaction of the particles, the scaffold has a porosity of 60%, interconnected, ideal for vascular and bone growth [27].

Finally, the choice of the animal model is also important. We used the rabbit skull, this model allowing the simultaneous use of 4 cylinders [22, 23, 28], and thus the testing of 4 simultaneous conditions on the same animal, a statistical and ethical argument.

Moreover, the morphology and bone metabolism are quite similar to humans [29], which makes it a clinically relevant model, used in ca. 80% of calvarial studies of vertical augmentation. The calvarial model may be clinically translated to a “one-wall defect”, such as a class IV defect in the jaw. The stringency of this model permits to accurately assess the vertical osteoconduction of the materials that are being evaluated. Herein, we advantageously employed the model’s characteristics to scrutinize the cellular and vascular mechanisms at play during the regeneration process.

We hope these amendments and explanations will prove satisfactory to the reviewer.

Reviewer 2 Report

Dear authors , congratulations for your work which give a good contribute to a trending topic in research field. Th paper is well written, it just need a little English review. Furrhemore, I suggest you to read and add to you references two paper inherent with your topic. The first one is : “Non-surgical periodontal treatment of peri-implant diseases with the adjunctive use of diode laser : preliminary clinical study” published in 2016. The other one is : “The skin rejuvenation associated tratment-Fraxel laser, Microbotox, and low G prime hyaluronic acid: preliminary results” published in 2019. 

I would like to see the paper once reviewed. Thank you for your work. 

Author Response

Dear authors , congratulations for your work which give a good contribute to a trending topic in research field. Th paper is well written, it just need a little English review. Furrhemore, I suggest you to read and add to you references two paper inherent with your topic. The first one is : “Non-surgical periodontal treatment of peri-implant diseases with the adjunctive use of diode laser : preliminary clinical study” published in 2016. The other one is : “The skin rejuvenation associated tratment-Fraxel laser, Microbotox, and low G prime hyaluronic acid: preliminary results” published in 2019. 

I would like to see the paper once reviewed. Thank you for your work.

> We thank the reviewer for her-his positive comments and assessment.

Concerning the suggested references, we have read these two articles thoroughly. In spite of the quality of this work that we appreciated, we regret that we could not find a reason for citing it as a reference to our purpose.

These two papers are related to a surgical treatment based on laser application.

The first one is about periimplantitis treatment, “The aim of this study was to compare conventional treatment of inflamed peri-implant tissues with conventional treatment together with diode laser application.”

The parameters that are evaluated are:

“At time T 0, before starting the treatment protocols, the first periodontal assessment was recorded, using a periodontal probe and a pressure of ≤0.15 N. All the clinical measurements were performed by the same operator, at T 0 and at T 1. Clinical data scores, including plaque index (PI), bleeding on probing (BoP), and pocket depth (PD), were collected:

 PI :     Presence or absence of plaque in four points around the implant—mesial, vestibular, distal, and lingual

 BoP :    Presence or absence of bleeding on probing in six points around the implant—distal-vestibular, vestibular, mesial-vestibular, mesial-lingual, lingual, and distal-lingual

 PD :    The distance in millimeters from the mucosal margin to the bottom of the pocket was taken at six points around each implant”

There is no mention of vascular cells, H cells, stem cells or biological mechanisms. Only clinical parameters are evaluated to measure the effectiveness of a laser treatment.

In this context, we do not know how to cite this reference or why to cite it.

The second study is a clinical case series in esthetic dermatology. It is aimed at demonstrating that a combination of hyaluronic acid injection + fraxel laser application + Microbotulinum toxin A injection “is better than single treatment and their association can boost the improvement of the skin texture and quality to rejuvenate patients’ face.”

Patients were followed up for 12 months with clinical checks at 1, 3, and 6 months. They were gave a clinical questionnaire (from 1 to 10) asking them to rate the satisfaction level

“The clinical result showed the following data:

  • In 98% (44 patients, 6 males, 38 females) of our patient population, a smoother skin surface, brighter, and more hydrated and elastic skin;
  • Sixty-eight percent of our patients (31 patients, 5 males, 26 females) showed less skin defects and staining as well as less small wrinkles, thanks to Fraxel laser treatment;
  • Ninety-eight percent (44 patients, 6 males, 38 females) showed tighter skin with less sebaceous gland secretion.”

The parameters we evaluated and the subject we worked on are completely absent from this second work and we do not understand how or why to cite it.

We hope that the reviewer will understand our position and will be satisfied by our decision, even if it contradicts her – his recommendation.

Reviewer 3 Report

The manuscript entitled “Identification of Type - H - like blood vessels in a dynamic and controlled model of osteogenesis in rabbit calvarium” may provide important information to the literature. However, a considered amount of issues should be addressed first:

The use of inadequate, scientifically imprecise terms, must be replaced in all sections of the manuscript, in addition to mistakes and mistyping

1.           Abstract is initiated by “correspondence”?

2.           Reference 12 should be replaced to the original author that described the finding.

3.           Authors quoted that “type-H vessels were discovered and defined as to their relative expression of CD31 and endomucin (EMCN) which is superior to that of other blood vessels present in the tissue.” How can one type of blood vessels be “superior” to others?

4.           What authors mean with “interlocutor”?

5.           “Here, we posited that....”Can authors explain “Here”?

6.           “The research current work” or The current research work ?

7.           Only 12 animals were used. It should not be mentioned animals used in other experiments of the research group.

8.           “intramuscular injection”, “iv”? Author should check International System of citation and standardize all along the manuscript.

9.           Some drugs' administration route are missing.

10.         “Hematoxylin-eosin staining was employed to topographically assess the section and for cell identification and discrimination.” What authors mean “assess”?

11.         What were the concentrations of the Probes? For how long? At what temperature? Tissue was pre-treated? There was post-treatment to eliminate probes inespecifically bounded and background signal?

12.         Most (if not all) in situ-hibridization methodology is missing.

13.         I suggest replacing: stemming.

14.         (Supplementary figure 1)?

15.         Where is the control group results in graphic C?

16.         “This relatively homogeneous tissue surrounded the particles, and consisted mainly of macrophages, neutrophils and fibroblasts.” Can authors identify macrophages exclusively on morphological basis on a H.E. stained slide?

17.         “fibroconjunctive tissue” or “fibroconnective tissue”

18.         “Remodeling zone: A progressive transformation from the osteogenic zone to an increasingly mature bone tissue closer to the bone bed was observed...”. Transformation is a term usually  applied in cancer research field. I suggest replace it.

19.         The Figures are too numerous and too small. Most are unreadable. I suggest a better selection and an adequate size.

20.         The “low”, “High” qualification of the probe sign is too subjective.

21.         “CD31 High / EMCN High” capillaries were observed statistically from 0 to ca. 3mm in height at 2 weeks...” What authors mean “observed statistically”?

22.         “In the end, this model is an extension of the bone system in which the tissue growth is controlled and may be easily visualized, a window on the bone metabolism.” What authors mean “In the end”?

23.         “we can say that we are in line with the expected results”. Explain.

24.         “In fine, this model is a dynamic model in which bone growth, and thus osteogenesis, is controlled in space and time.” Explain “In fine”.

25.         “Finally, we developed a technically and statistically robust analytical tool that enabled...”I suggest deleting, It is too pretentious to a scientific writing.

26.         “The parallelism”. I suggest replacing.  

Author Response

The manuscript entitled “Identification of Type - H - like blood vessels in a dynamic and controlled model of osteogenesis in rabbit calvarium” may provide important information to the literature. However, a considered amount of issues should be addressed first:

The use of inadequate, scientifically imprecise terms, must be replaced in all sections of the manuscript, in addition to mistakes and mistyping

> We acknowledge reviewer 3 for her – his positive evaluation and the constructive remarks she-he brings us. This will help improving the quality of our work. We hope that the changes we have made in the manuscript and which are detailed below will answer all her-his remarks and bring her-him complete satisfaction.

In addition, the text was thoroughly reviewed and edited.

  1. Abstract is initiated by “correspondence”?

Sorry for this mistake, “correspondence” was replaced by “abstract”

  1. Reference 12 should be replaced to the original author that described the finding.

The reference was modified. We selected a paper from Garlanda and Dejana: Heterogeneity of endothelial cells. Specific markers. Arterioscler Thromb Vasc Biol. 1997; 17:1193–1202.

  1. Authors quoted that “type-H vessels were discovered and defined as to their relative expression of CD31 and endomucin (EMCN) which is superior to that of other blood vessels present in the tissue.” How can one type of blood vessels be “superior” to others?

We understand our formulation was unclear and thank the reviewer. The text was modified as follows and may be clearer to readers:

“Recently, type-H vessels were discovered. Their main characteristic is a level of CD31 and endomucin (EMCN) expression which is higher to that of other blood vessels. They are defined and analyzed on the basis of CD31 and EMCN expression.”

  1. What authors mean with “interlocutor”?

We understand that the term interlocutor is inappropriate. We modified the text as follws and hope it will be clearer:

"In osteogenic niches from the bone vascular growth front and the endosteum, Type-H vessels were identified together with mesenchymal stem cells and osteoprogenitors (OP)"

  1. “Here, we posited that....”Can authors explain “Here”?

The sentence was modified:

"In the present study, we postulated that…"

  1. “The research current work” or The current research work ?

The sentence was modified:

"The research work we describe here is part of a…"

  1. Only 12 animals were used. It should not be mentioned animals used in other experiments of the research group.

We thank the reviewer for this remark. However, we would like to leave the description of the number of animals as it is. Indeed, to understand our allocation table, one must absolutely be aware that there were more than 12 animals in total. If we had used only 12 animals, it would have been impossible to have 6 biological replicates from 6 different animals at 2 time-points, which is an important point of statistical relevance in our study. Therefore, we decided to leave this description as is and we hope that the reviewer will accept this in view of the explanations we give here.

  1. “intramuscular injection”, “iv”? Author should check International System of citation and standardize all along the manuscript.
  2. Some drugs' administration route are missing.

We thank the reviewer to have highlighting these mistakes. We amended the text as follows:

Surgical procedures were performed as previously described by our group [22, 23].

Briefly, animals were left for one-week acclimation in the facility and received a prophylactic broad-spectrum antibiotic coverage (2h preoperative, 3 days postoperative, enrofloxacine Baytril 10%, 5-10 mg/kg PO, Bayer, Leverkusen, DE). On the day of the surgeries, they were pre-anesthetized by an injection of ketamine (25 mg/kg, 50mg/ml, 0.5ml/kg IM, Pfizer, NY, US) plus xylazin (3mg/kg, 20 mg/ml, 0.15ml/kg IM, Bayer). Deep anesthesia was induced by propofol 2% IV (Braun, Sempach, CH) and maintained with sevoflurane 3% (Abbvie, Chicago, IL, US) in pure oxygen. A remifentanil perfusion (Bichsel, Unterseen, CH) (ear vein, 0.008-0.5um/kg/min, 5 g/ml IV) ensured analgesia during the procedure. A midsagittal incision and periosteum elevation were then performed before screwing of 4 PEEK cylinders and drilling of 5 intramedullary holes (diameter 0.8 mm, depth ca. 1 mm) in the bone bed of each cylinder. After filling the cylinders with bone substitutes and closing (Fig. 1B), the surgical site was sutured with intermittent non-resorbable sutures (Prolene 4.0, Ethicon, Somerville, NJ, US). Animal received buprenorphine hydrochloride (Reckitt Benckiser, Slough, UK), every 6h and up to 3 days (0.02 mg/kg, 0.03 mg/ml, 0.67 ml/kg SC) as postoperative analgesia. Suture were removed after ca. 10 days.

  1. “Hematoxylin-eosin staining was employed to topographically assess the section and for cell identification and discrimination.” What authors mean “assess”?

The text was modified as follows in response to the reviewer’s remark:

"Hematoxylin-eosin staining was employed to describe the general aspect of the section, of tissue and scaffold structure and for cell identification and discrimination."

  1. What were the concentrations of the Probes? For how long? At what temperature? Tissue was pre-treated? There was post-treatment to eliminate probes inespecifically bounded and background signal?
  2. Most (if not all) in situ-hibridization methodology is missing.

We thank the reviewer and understand her-his needs for a detailed protocol. We amended the text as follows, page 4, MM section:

"Briefly, sections were first deparaffinized, heated at 85°C for 10 minutes and digested with a protease solution (dil. 1/100) for 20 minutes at 40°C. Once pretreated, the sections were re-fixed in 10% neutral buffer formalin for 5 minutes at room temperature. Probes sets were then applied for 3 hours at 40°C (CD31 (PECAM type 1) - Osterix (Osx type 6) and CD31 (PECAM type 1) - Osterix (Osx type 6)) to a final dilution of 1/80. As a negative control, probe set diluent was applied without any probe. Slices were rinsed vigorously several times before signal pre-amplifier and amplifier hybridization at 40°C. Label probe (type 6) hybridization was then performed (dil. 1/1000, 15 minutes at room temperature) before the fast-blue substrate was added (darkness, room temperature for 30 minutes). After rinsing, Label probe (type 1) hybridization was operated (dil. 1/1000, 15 minutes at 40°C) and the fast-red substrate was added for 60 minutes at room temperature. Slices were finally coverslipped."

  1. I suggest replacing: stemming.

The sham sites did not support bone growth in height (Fig.2C), yet very little amounts of newly formed bone were observed at the bone bed surface, coming from the intramedullary holes healing (Supplementary figure 1).

  1. (Supplementary figure 1)?

The figure was tagged as scheme 1 by mistake (p17, appendix)- We modified the caption and titled “supplementary figure 1”

  1. Where is the control group results in graphic C?

We added the controls on the graphic C from figure 2 and updated the corresponding caption.

  1. “This relatively homogeneous tissue surrounded the particles, and consisted mainly of macrophages, neutrophils and fibroblasts.” Can authors identify macrophages exclusively on morphological basis on a H.E. stained slide?

Reviewer is true, H.E is not sufficient to define macrophages. A CD68 staining for example is needed to conclude definitively.

The text was modified as follows:

"This relatively homogeneous tissue surrounded the particles, and consisted mainly of mononuclear- and some polymorphonuclear- cells."

  1. “fibroconjunctive tissue” or “fibroconnective tissue”

We thank the reviewer for her-his remark. The mistake was corrected and fibroconjunctive was replaced by fibroconnective.

  1. “Remodeling zone: A progressive transformation from the osteogenic zone to an increasingly mature bone tissue closer to the bone bed was observed...”. Transformation is a term usually applied in cancer research field. I suggest replace it.

The text was modified as follows:

"The granulation tissue evolved progressively into a highly vascularized osteoid tissue (Fig. 2 A-B)."

  1. The Figures are too numerous and too small. Most are unreadable. I suggest a better selection and an adequate size.

We thank reviewer 2 for her – his remark. We modified the figures as follows:

- Figure 1 was not modified.

- Old Figure 2 was split into two figures:

> new figure 2 with histological pictures and histomorphometry. The pictures were enlarged, and the graphic C was completed with controls.

> New figure 3, with the same IF pictures enlarged

- Old Figure 3  was modified and tagged as figure 4. IF picture in A. showing the cylinder entire content was deleted, the capillaries projection was conserved and enlarged. Panels B. and C. were not modified.

- Old figure 4 was not modified and was tagged as figure 5

- Old figure 5 was tagged as figure 6 and modified. IF pictures from the cylinder entire content were deleted at 2 and 4 weeks. Only the capillaries projections were left in A. Pictures in B. were not modified but the whole figure was enlarged.

Captures and numeration were modified and adapted to this new configuration.

  1. The “low”, “High” qualification of the probe sign is too subjective.

The technique we developed with the help of Thermofisher (RNA view) was optimized so that highly qualitative and semi-quantitative results may be obtained. Briefly, small oligos with high specificity are used, combined to a branched-DNA signal amplification. It results in a high specificity, low background, and high signal to noise ratio.

mRNA may thus be analyzed with a single copy sensitivity to such an extent that a single colored dot corresponds to a single mRNA copy.

Based on the data obtained with this technique, we developed an original method of semi-quantification with our local bioengineer specialized in bioimaging. Since a colored dot corresponded to a mRNA copy, related pixels were counted within areas that we delimited with respect to capillaries’ morphology. In the end, the relative expression of the different targets was then calculated with a high sensitivity.

To discriminate between CD31-EMCN high-high and low-low, we used the scatter plots representing the capillaries as to their relative expression of CD31 and EMCN. Each dot was a capillary. Once they were expressing the same relative proportion of CD31-EMCN, these capillaries appeared at the same location on dot plots, so that the capillaries could be also classified thanks to their density in a specific position on the dot-plots. A color code was used to tag this capillaries’ density. We used this color code to discriminate between CD31-EMCN high-high and low-low capillaries, and placed our threshold on each dot plot at the at the limit of the denser core, i.e. the red-orange-yellow capillaries’ population. Thus, most of the capillary density was in the “CD31 Low / EMCN Low” category, which corresponds to physiological situation where H-capillaries are rare.

To clarify our manuscript, the text was modified as follows in the MM section p5:

"Thresholds to define high versus low CD31 and EMCN relative areas in capillaries were user defined per sample.They were set in order to have most of the capillary density in the “CD31 Low / EMCN Low” category. Practically, a cross delimiting the different categories of capillaries with respect to their CD31-EMCN expression was placed on scatter plots. Capillaries density was color coded, and the cross was always placed at the limit of the denser core, i.e. the red-orange-yellow capillaries’ population."

  1.  “CD31 High / EMCN High” capillaries were observed statistically from 0 to ca. 3mm in height at 2 weeks...” What authors mean “observed statistically”?

Representations shown in fig 5 B-C-D come from a statistical repartition. By extension, we used the term “observed stastistically”. We agree that this formulation is inappropriate and the term statistically was deleted:

"CD31 High / EMCN High” capillaries were observed statistically from 0 to ca. 3mm in height at 2 weeks."

  1. “In the end, this model is an extension of the bone system in which the tissue growth is controlled and may be easily visualized, a window on the bone metabolism.” What authors mean “In the end”?

The text was modified as follows:

"This model may therefore be considered as an extension of the bone system in which the tissue growth is controlled and may be easily visualized, a window on the bone metabolism."

  1. “we can say that we are in line with the expected results”. Explain.

We thank the reviewer for her-his remark and gave some more explanation:

"These measures fit with data form other studies [26] and from our own experience [22] where ca. 12% bone filling over the entire scaffold were reached after 12 weeks. With 50% filling at 4 weeks and about 5% new bone, we can extrapolate to 10-12% at 12 weeks with a complete cylinder filling, and consider the bone growth process in accordance with the model we used."

  1. “In fine, this model is a dynamic model in which bone growth, and thus osteogenesis, is controlled in space and time.” Explain “In fine”.

The text was amended as follows:

"Considering all these observations, this model is a dynamic model in which bone growth, and thus osteogenesis, is controlled in space and time."

  1. “Finally, we developed a technically and statistically robust analytical tool that enabled...”I suggest deleting, It is too pretentious to a scientific writing.

We modified the sentence as proposed:

"Finally, we developed an analytical tool that enabled us to demonstrate that the capillaries identified in our biopsies express CD31 and EMCN."

  1. “The parallelism”. I suggest replacing.

The test was modified as follows:

"The resemblance between the endochondral and intramembranous systems does not end with this observation. Indeed, in the endonchondral system, it was shown that type-H capillaries decrease with age, especially in embryonic bone development, as ossification increases [11, 13]."

Reviewer 4 Report

This is an interesting manuscript. However, the authors' tools for studying the relevant results are too homogeneous, relying mostly on qualitative confocal and local staining tests, which suffer from a lack of confidence in the data. Therefore, the authors need to add quantitative tests such as WB and PCR to further enhance the reliability and reproducibility of the results.

In addition, the authors need to further explain the relevance of their research to materials science in order to better meet the objectives of materials.

Major concerns:
1. The English should be improved.
2. The author should claim the relationship of the results with human.

Author Response

"This is an interesting manuscript. However, the authors' tools for studying the relevant results are too homogeneous, relying mostly on qualitative confocal and local staining tests, which suffer from a lack of confidence in the data. Therefore, the authors need to add quantitative tests such as WB and PCR to further enhance the reliability and reproducibility of the results.”

> We thank reviewer 4 for the quite positive assessment of our work.

As the study was based on animal sacrifices, our priority was focused on 3R rules’ respect.

We were inspired by the foundingl works from Ramasamy et al in 2014  who were the first to describe H-capillaries on the basis of immunofluorence and FACS analysis (Nature (2014), 507(7492), 323-8.).

As we were limited with the number of animals, we decided to design the experiment by using a biopsy treatment that could serve the most techniques as possible. By fixing the biopsies in formalin and embed it in paraffin, we were able to perform not less than 5 different staining that is HE, Masson, CD31, EMCN and osterixFlow cytometry analysis would have needed the sacrifice of a larger number of animals, and we then hypothesized that mRNA hybridization with a new generation technique would be powerful enough to give our results a robust statistical analysis. 

We understand that reviewer is not fully convinced by mRNA hybridization to perform semi-quantitative analysis and suggest that RT-PCR and-or WB should be used in addition.

This is true that traditional FISH techniques are more qualitative than quantitative. The use of large oligonucleotide as target-probes combined to an insufficient signal amplification lead to a low sensitivity and a high background.

The technique we developed with the help of Thermofisher (RNA view) was optimized so that highly qualitative and semi-quantitative results may be obtained. Briefly, small oligos with high specificity are used, combined to a branched-DNA signal amplification. It results in a high specificity, low background, and high signal to noise ratio.

mRNA may thus be analyzed with a single copy sensitivity to such an extent that a single colored dot corresponds to a single mRNA copy.

Based on the data obtained with this technique, we developed an original method of semi-quantification with our local bioengineer specialized in bioimaging. Since a colored dot corresponded to a mRNA copy, related pixels were counted within areas that we delimited with respect to capillaries’ morphology. In the end, the relative expression of the different targets was then calculated with a high sensitivity.

The technique that we developed is therefore sensitive and semi-quantitative, and we are confident that it is statistically and scientifically supporting our purpose.

Adding other quantitative methods would be undoubtedly a plus and if we could, we would do it. However, our biopsies were almost completely operated and the little that remains is too old to be treated by using molecular or protein analysis like WB and qPCR. This is an advantage from our RNA view method that can work on partially degraded mRNA when qPCR can’t.

On the other hand, qPCR and WB are based on complete mRNA or protein extractions. In other words, all the cell types from our biopsies (complete biopsy or layers of 1mm height) would have been mixed. CD31 is not only expressed by endothelial cells, but also by macrophages, granulocytes, osteoclasts, platelets, etc (J Cell Sci (2013) 126 (11): 2343–2352), so as EMCN which is expressed by endothelial cells plus hematopoietic stem cells and adipocytes (Blood Adv. (2018) Jul 10; 2(13): 1628–1632). These cells are present on our biopsies and qPCR or WB would have been not selective enough to perform a qualitative analysis.

FACS would have been definitely the solution to reinforce our results. In effect, cell polulations’ seletion on the basis of CD31 and EMCN co-expression is possible. But as discussed above, it was not plan in our initial design due to 3R respect. We are confident that the analysis we performed is semi-quantitative and hope that our strategy will convince reviewer 4 in view of the new arguments we have brought.

In addition, the authors need to further explain the relevance of their research to materials science in order to better meet the objectives of materials.

Many thanks for your remark. This is true that this paper is essentially a work on the biological part of the bone regeneration. It has been submitted within the special issue “ Improving Bone Tissue Engineering and Regeneration at the Biological and Biomaterials Level” which allows to treat biological subjects in line with the improvement of bone grafting materials.

We believe that this paper which is based on a traditional bone augmentation model designed to study new materials may help materials readers to better understand bone biology and improve their materials developments.

In this context, we amended our introduction as follows:

A better understanding of bone regeneration is a key for improving bone scaffolds’ materials and designs. 

Here, we postulated that type-H-like capillaries may be present at sites of intramembranous bone development and participate in the control of osteogenesis. With the aim to control and visualize the entire process of bone growth over time, we used a model of calvarial bone augmentation in the rabbit. The formation of type-H-like capillaries was tracked over time and correlated with the presence of osteogenic precursors using systematic semi-quantitative mRNA hybridization for CD31, EMCN and Osterix.”

The English should be improved.

The text was thoroughly reviewed and edited.

The author should claim the relationship of the results with human.

We thank the reviewer for her-his remark. The model was already discussed that way in the discussion, first paragraph. Since it appeared that the relation to human clinics was not enough explained, we added this text highlighted to the first paragraph of the discussion:

In the bone regeneration model employed herein, several identical cylinders are screwed onto the skull of an animal and filled with a scaffold [22, 23]. The scaffold is necessary for bone growth, as it triggers a vertical "ectopic" osteogenesis that extends above the native calvarial bone plateau. If the regenerative cells are not guided and supported, then no bone tissue is naturally synthesized. The choice of the scaffold is therefore crucial, both in its biochemical and architectural nature. In the end, this model is an extension of the bone system in which the tissue growth is controlled and may be easily visualized, a window on the bone metabolism.

In the present study, the Shams were filled with coagulated blood only. Apart from the healing of intramedullary perforations made to attract osteogenic precursors and vascularization, no bone synthesis was observed. This is in accordance with other studies in which this model of ectopic augmentation was used [22,23,25,26]. The scaffold employed consisted in 0.25-1 mm bovine bone particles that represent a gold standard in orofacial bone regeneration for several decades. After compaction of the particles, the scaffold has a porosity of 60%, interconnected, ideal for vascular and bone growth [27].

Finally, the choice of the animal model is also important. We used the rabbit skull, this model allowing the simultaneous use of 4 cylinders [22, 23, 28], and thus the testing of 4 simultaneous conditions on the same animal, a statistical and ethical argument.

Moreover, the morphology and bone metabolism are quite similar to humans [29], which makes it a clinically relevant model, used in ca. 80% of calvarial studies of vertical augmentation. The calvarial model may be clinically translated to a “one-wall defect”, such as a class IV defect in the jaw. The stringency of this model permits to accurately assess the vertical osteoconduction of the materials that are being evaluated. Herein, we advantageously employed the model’s characteristics to scrutinize the cellular and vascular mechanisms at play during the regeneration process.”

We hope these amendments and explanations will prove satisfactory to the reviewer.

Round 2

Reviewer 3 Report

Manuscript is ready for publication now.

Reviewer 4 Report

I appreciate the authors' revisions and explanations in all areas. However, the data presented in this paper hardly support the conclusions expressed so far. The authors have tried to convince the reviewers of the validity of the data in their response, but the poor characterization is not convincing. Further the relationship between the manuscript and the material is debatable.